# Can Data Assimilation Improve Short-Term Prediction of Land Surface Variables?

**Yingze Tian [1], Tongren Xu [1,*], Fei Chen [2], Xinlei He [1] and Shi Li [1]**

[1] State Key Laboratory of Earth Surface Processes and Resource Ecology, School of Natural Resource, Faculty of Geographical Science, Beijing Normal University, Beijing 100875, China

[2] National Center for Atmospheric Research, Boulder, CO 80301, USA

\* Correspondence: xutr@bnu.edu.cn

**Abstract:** Data assimilation methods have been used to improve the performances of land surface models by integrating remote sensing and in situ measurements. However, the impact of data assimilation on improving the forecast of land surface variables has not been well studied, which is essential for weather and hydrology forecasting. In this study, a multi-pass land data assimilation scheme (MLDAS) based on the Noah-MP model was used to predict short-term land surface variables (e.g., sensible heat fluxes (H), latent heat fluxes (LE), and surface soil moisture (SM)) by jointly assimilating soil moisture, leaf area index (LAI) and solar-induced chlorophyll fluorescence (SIF). The test was conducted at the Mead site during the growing season (1 May to 30 September) in 2003, 2004, and 2005. Four assimilation-prediction scenarios (assimilating for 15 days, 45 days, 75 days, and 105 days from 1 May, then predicting one future month) are adapted to evaluate the influence of assimilation on subsequent prediction against Noah-MP open-loop simulation (OL). On average, MLDAS produces 28.65%, 27.79%, and 19.15% lower root square deviations (RMSD) for daily H, LE, and SM prediction compared to open-loop run, respectively. The influence of assimilation on prediction can reach around 60 days and 100 days for H (LE) and SM, respectively. Our findings indicate that data assimilation can improve the accuracy of land surface variables in a short-term prediction period.

**Keywords:** sensible heat flux; latent heat flux; short-term prediction; ensemble Kalman filter; Noah-MP

## 1. Introduction

It is critical to monitor and predict land surface variables related to vegetation dynamics, water and carbon fluxes, which are essential for better understanding complex land–atmosphere interactions and climate change [1–3]. Process-based land surface models (LSMs) make it possible to simulate land surface variables and represent complex land surface processes [4]. A series of models have been developed to study land and lower atmosphere conditions. Among them, the Noah with multi-parameterization model (Noah-MP model) significantly improves the simulations of land surface vegetation and heat fluxes with its advantages of using multiple optional parameterization schemes to interpret the differences in modeling simulations [5].

While LSMs can provide temporally and spatially continuous simulations of land surface variables, there are still substantial uncertainties caused by initial conditions, forcing data, interactions between land surface and atmosphere, and model physics within LSMs, leading to significant differences between simulation and reality [6]. Data assimilation (DA) methods that can merge observational information into process-based LSMs can constrain parameters and optimize the simulations of land surface variables. Most data assimilation schemes are developed with variational-based or ensemble-based methods to improve the performance of LSMs [4–8]. Based on the assimilation theories, various observational variables such as land surface temperature (LST), leaf area index (LAI), and soil moisture

(SM) observations have been assimilated into LSMs, improving the simulations of land surface variables from LSMs. For instance, assimilation of SM improves the estimation of evapotranspiration [9,10], and assimilation of LAI optimizes estimates of carbon fluxes of gross primary production (GPP) and net ecosystem exchange (NEE) [11–13].

It is notable that these traditional data assimilation frameworks are built with a single or dual observational variable assimilated into theprocess-based models [11,14,15]. Albergel et al. [16] found that root zone soil moisture (RZSM) can benefit from the assimilation of SM and LAI, which indicates that assimilation of multi-variables at once may improve the performance of land surface variable simulations. Recently, a multi-pass land data assimilation scheme (MLDAS) has been proposed, which can assimilate different observational variables into LSMs to optimize simulations of land surface variables all at once [17]. As different observed variables are jointly assimilated, MLDAS potentially improves the capability of LSM simulations.

Although some data assimilation schemes have been constructed within LSMs to better simulate land surface variables, few studies are focused on the potential of improvement of predictivity skills brought by data assimilation. It is believed that uncertainties of land surface forecast mainly arise from initial conditions (ICs), meteorological forcing, and errors in model structure, and initial conditions play an important role among them. Wood and Lettenmaier [7] evaluated the relative importance of initial conditions and meteorological forcing in hydrologic forecast and discovered that initial conditions dominated the forecast in some cases. Sawada et al. [18] assimilated microwave brightness temperature into a land surface model for better monitoring and forecast of agricultural drought. The results showed that data assimilation especially sequential data assimilation may provide more accurate initial conditions that lead to better LAI and SM forecasts, which were likely to benefit agricultural drought monitoring. Quantities of data assimilation experiments have proved that accurate initial conditions are crucial for prediction, but the observational data are assimilated into the LSM separately in those experiments. It is a topic of interest to test whether assimilating several land surface variables into LSM jointly would improve prediction of LSMs.

The objective of this study is to test whether Noah-MP LSM coupled with MLDAS (Noah-MP-MLDAS) can better predict land surface variables such as LAI, SM, sensible heat flux (H) and latent heat flux (LE) compared to Noah-MP open-loop simulations (OL) (simulations of Noah-MP without assimilation). Assimilating for 15 days, 45 days, 75 days, and 105 days from May 1st, four sets of data assimilation experiments (four scenarios) were designed and conducted at an AmeriFlux crop site, namely Mead (Mead, NE, USA) during the growing season (1 May to 1 October) in 2003, 2004 and 2005, respectively. The SM, H, and LE assimilation outputs for the growing season served as benchmarks for evaluating the predictability of MLDAS and open-loop run (OL), the duration of impact on prediction in each scenario was further evaluated. This study demonstrates how data assimilation can improve the prediction capability of LSM and is expected to be useful for scientific research on LSM prediction and agricultural monitoring.

## 2. Materials and Methods

### 2.1. Data

The assimilation-prediction experiment was conducted using MLDAS at the AmeriFlux Mead (US-Ne1) site, which is located at the University of Nebraska Agricultural Research and Development Center. There are 3 sites (US-Ne1, US-Ne2, and US-Ne3) over Mead which are within 1.6 km of each other. Site 1 (US-Ne1), used in our study, is an irrigated site covered by corn [19].

Meteorological forcing data of the Mead site, including air temperature, pressure, humidity, precipitation, wind speed, incoming shortwave and longwave radiation used to drive Noah-MP, were downloaded from AmeriFlux (http://ameriflux.lbl.gov/, accessed on 7 October 2021). LAI and SM were also acquired from the AmeriFlux database. SM was measured at depths of 0.02, 0.05, and 0.1 m, which was averaged to be consistent with

the topsoil layer of Noah-MP (0.1m) [17]. The remotely-sensed solar-induced chlorophyll fluorescence (SIF) products with the spatiotemporal resolution of 5 km and 4 days were available via https://osf.io/8xqy6/?view_only=4d14a64b7be644d-39564c82a371e1d20 (accessed on 1 November 2021) [20].

## 2.2. Noah-MP Model

Noah-MP is enhanced from Noah LSM [21] with expanded and multi-parameterization options [4,5]. A two-source energy balance module, short-term dynamic vegetation growth model, modified two-stream radiation transfer scheme, dynamic groundwater component, and multilayer snowpack are included in Noah-MP LSM. The multi-parameterization options make it possible for users to try combinations of parameters in different modules to optimize the model performance, such as leaf dynamics, surface-layer turbulence treatment, canopy stomatal resistance, runoff, and groundwater [5]. A crop module has also been added to Noah-MP LSM, improving the simulations of vegetation dynamics and turbulent fluxes for corn and soybean by using several agricultural management data, such as growing degree days (GDDs) and planting/harvesting dates [8,9,22].

## 2.3. Multi-Pass Land Data Assimilation Scheme

A multi-pass land data assimilation scheme (MLDAS) has been proposed based on the Noah-MP model [4,5] and ensemble Kalman filter (EnKF) method. Unlike other data assimilation (DA) methods, MLDAS can assimilate multisource remotely sensed or ground-measured observations in multiple passes to optimize parameters and state variables simultaneously. Currently, four passes have been built to assimilate LAI (pass I–pass II), SM (pass III), and SIF (pass IV) to update specific leaf area (SLA), leaf biomass (LFMASS), SM and maximum carboxylation rate at a reference temperature of 25 °C (Vcmax25) in MLDAS successfully [17]. Pass I to pass II are designed to update simulation on plant productivity, ecological behavior and plant growing rates, which are key factors for vegetation status and evapotranspiration. SLA is defined as ratio of LAI to LFMASS, so LAI can be assimilated in two passes to update these two parameters jointly. In pass I, LAI is assimilated to update the constant parameter SLA which is equivalent to BIO2LAI in Noah-MP. In pass II, state variable LFMASS is updated. SM is updated in pass III directly. Through assimilating SIF in pass IV, we can optimize Vcmax25 to better control carbon assimilation processes and represent plant productivity [23].

In our study, four passes were opened to jointly assimilate LAI, SM and SIF. LAI observations will be assimilated when available, and SM and SIF observations will be assimilated when available at daily and four-day intervals, respectively.

## 2.4. Experiments Setup

In this study, the MLDAS system was used to test the impact of data assimilation on land surface variable prediction. Using the same atmospheric forcing data, the MLDAS predictions (red arrows in Figure 1) were compared with simulations of Noah-MP OL (without assimilation, blue arrow in Figure 1) to evaluate the influence of assimilation on predictions. Four assimilation-prediction scenarios (assimilating for 15 days, 45 days, 75 days, and 105 days from 1 May, and then predicting until 1 October) were adapted during the growing season (1 May to 30 September) in 2003, 2004, and 2005, respectively (as shown in Figure 1). Furthermore, considering that the differences between OL and MLDAS are significant in the first month of prediction, the first 30 days of the prediction period of each scenario (15 May–15 June, 15 June–15 July, 15 July–15 August, and 15 August–15 September for scenario I, II, III, and IV, respectively) were used for statistical analysis. This part corresponds to the red-dashed lines in Figure 1. Notably, the main goal of this study was to assess to what extent data assimilation could influence the prediction of land surface variables, so we took the output of assimilation for the whole growing season (1 May to 30 September) as the "benchmarks" that contain information from both models and observations considering the uncertainties in raw observations. The "benchmarks"

corresponds to the green arrow in Figure 1 and will be further compared with the prediction of MLDAS and OL.

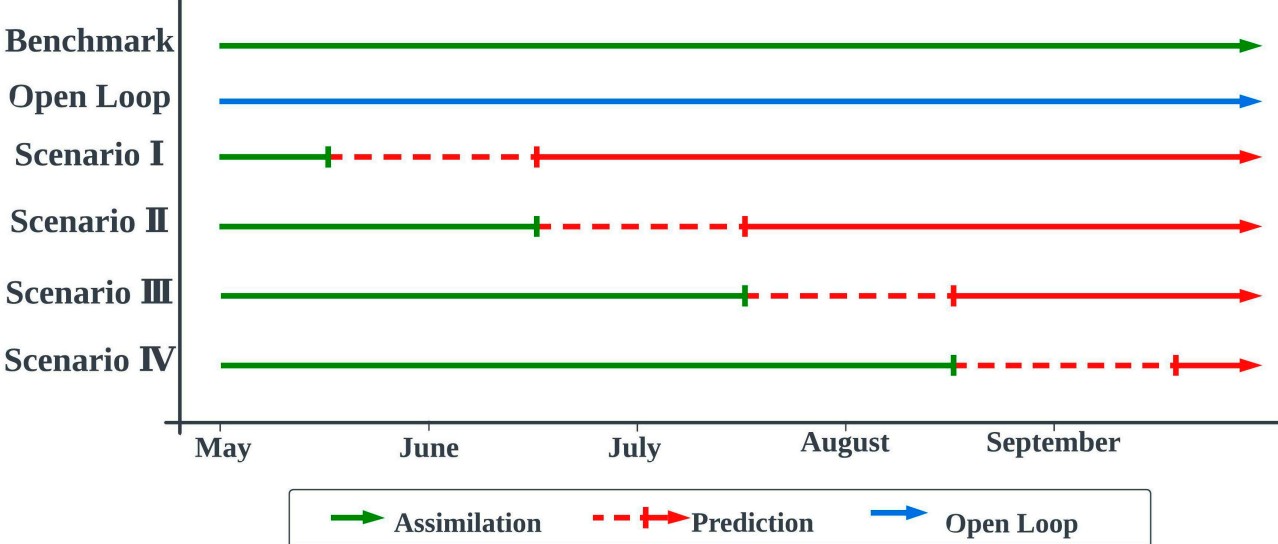

**Figure 1.** Diagram for four assimilation scenarios. The green arrow represents benchmarks with data assimilation; the blue arrow represents Noah-MP Open-Loop run (without data assimilation); the red arrows represent the prediction of LAI, SM, H, or LE; and the dashed lines in red arrows are the first-month prediction used for statistical analysis.

Here, normalized information contribution ($NIC_{RMSD}$) was adapted to quantify the differences between MLDAS and OL, which was defined by Equation (1) [24]. $RMSD_{Analysis}$ ($RMSD_{Model}$) refers to RMSD between "benchmarks" and MLDAS (OL). A positive $NIC_{RMSD}$ indicates the prediction of MLDAS is improved compared with OL. A negative $NIC_{RMSD}$ means degradation. Besides, the relative difference (*RD*) between OL and MLDAS defined by Equation (2) was used to show how long the influence of assimilation exists during the prediction period. *RD* in the equation is relative difference, $LSV_{Model}$ and $LSV_{Analysis}$ are simulations of land surface variables (LSV) from OL and MLDAS, respectively.

$$NIC_{RMSD} = \frac{RMSD_{Model} - RMSD_{Analysis}}{RMSD_{Model}} \times 100 \tag{1}$$

$$RD = \frac{\left| LSV_{Model} - LSV_{Analysis} \right|}{LSV_{Model}} \times 100 \tag{2}$$

## 3. Results

### 3.1. Impacts of Assimilation on Land Surface Variables Prediction

The output time step of MLDAS is 30 min, so H, LE, and SM predictions were daytime-averaged (09:00–18:00, local time) for further estimates within all assimilation-prediction scenarios. As with each assimilation scenario, a short-term prediction (forecasting lead time of 30 days) for LAI, SM, H, and LE was conducted separately in MLDAS and OL. $NIC_{RMSD}$ was used to show the advances in land surface variables prediction by MLDAS compared with Noah-MP OL.

#### 3.1.1. Leaf Area Index

Figure 2 shows the prediction of LAI in MLDAS and OL at the Mead site from 2003 to 2005. Since LAI observations are limited, the ground-measured LAI was used as the references in this section, and $NIC_{RMSD}$ of LAI was not calculated herein. As shown, MLDAS performs better than OL in most scenarios compared to ground-measured LAI. In

general, LAI is tended to be overestimated, especially during DOY 180–220 in OL. MLDAS corrects the overestimation and produces predictions that can better fit LAI observations, suggesting that assimilation can bring better prediction skills to MLDAS. However, as Figure 2 shows, the prediction skill of MLDAS is worsened in scenario I assimilating for 15 days in 2004 and 2005. This is because only one set of LAI observations with relatively low values was assimilated into the Noah-MP model, bringing uncertainties to future prediction. In contrast, longer assimilation then prediction generally leads to better predictions of LAI peaks, which are especially better captured in scenario III from 2003 to 2005. It is also notable that the correction in MLDAS is most obvious in midsummer, when LAI is at peak. Correction brought by MLDAS is no longer evident compared with OL after midsummer (scenario IV).

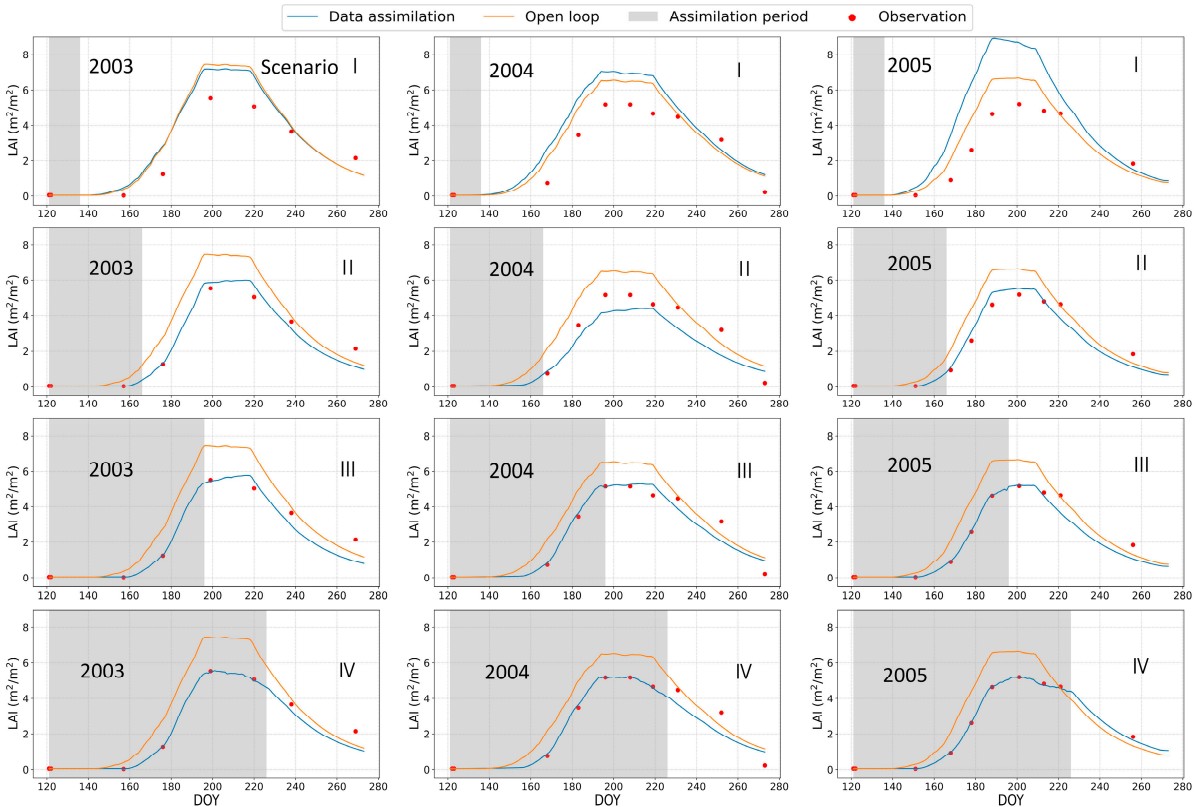

**Figure 2.** Leaf area index simulation and prediction from OL and MLDAS in 2003–2005. Results of four assimilation scenarios with different assimilation periods are given. Columns represent different assimilation scenarios in each year. The blue line in each subplot represents MLDAS simulation and prediction; the orange line in each subplot represents OL simulation; the red scatters represent LAI observations; and the shaded regions are the assimilation period, which starts from 1 May and lasts for 15 days, 45 days, 75 days, and 105 days for scenarios I, II, III, and IV, respectively.

### 3.1.2. Soil Moisture

The statistics of $RMSD_{Analysis}$, $RMSD_{Model}$, and $NIC_{RMSD}$ at the Mead site from 2003 to 2005 are shown in Table 1. As is shown, 9 out of 12 experiments experience improvement in the prediction from OL to MLDAS. The maximum relative difference is up to 60.41% in scenario IV in 2003. There are three sets of experiments that appear to be a degradation of predicting performance after assimilation. The $NIC_{RMSD}$ is negative in scenarios I and II from 2003 to 2005, indicating that SM prediction improvements are unstable with a short data assimilation period (i.e., 15 or 45 days). The $NIC_{RMSD}$ of scenarios III and IV generally increases in all 3 years, suggesting that forecast performance is better with the extended assimilation period.

**Table 1.** $NIC_{RMSD}$ of soil moisture in different assimilation scenarios (I–IV) in 2003, 2004, and 2005. Positive $NIC_{RMSD}$ numbers are marked in bold.

| Year | 2003 | | | 2004 | | | 2005 | | |
|---|---|---|---|---|---|---|---|---|---|
| | $RMSD_{Model}$ (m³/m³) | $RMSD_{Analysis}$ (m³/m³) | $NIC_{RMSD}$ (%) | $RMSD_{Model}$ (m³/m³) | $RMSD_{Analysis}$ (m³/m³) | $NIC_{RMSD}$ (%) | $RMSD_{Model}$ (m³/m³) | $RMSD_{Analysis}$ (m³/m³) | $NIC_{RMSD}$ (%) |
| Scenario I | 0.0102 | 0.0106 | −3.87 | 0.0233 | 0.0115 | **50.51** | 0.0084 | 0.0106 | −26.50 |
| Scenario II | 0.0138 | 0.0136 | **1.23** | 0.0121 | 0.0136 | −12.08 | 0.0469 | 0.0402 | **14.26** |
| Scenario III | 0.0386 | 0.0245 | **36.35** | 0.0224 | 0.0201 | **10.10** | 0.0251 | 0.0225 | **10.15** |
| Scenario IV | 0.0540 | 0.0214 | **60.41** | 0.0244 | 0.0166 | **31.93** | 0.0411 | 0.0227 | **44.83** |

The detailed impacts of data assimilation on SM prediction are shown in Figure 3. As is shown, Noah-MP OL tends to overestimate SM at the Mead site from 2003 to 2005. For assimilation-prediction scenarios I and II, the data assimilation performs similarly to the OL. However, with longer data assimilation periods (scenarios III and IV), data assimilation can significantly improve SM prediction compared to OL. For assimilation-prediction scenario IV (assimilation of 105 days), the prediction of SM can reach even the maximum and minimum benchmark values for the whole future month, indicating that the data assimilation significantly impacts the prediction of future SM.

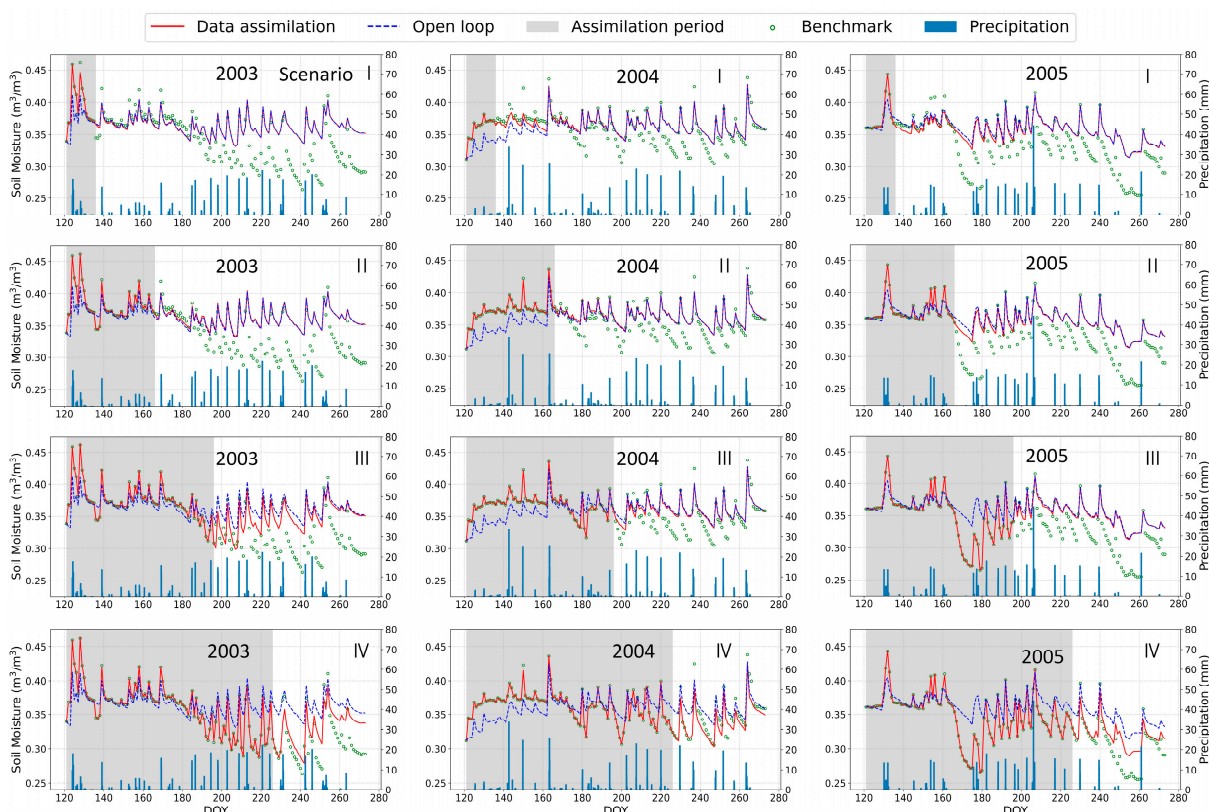

**Figure 3.** Soil moisture simulation and prediction from OL and MLDAS in 2003–2005. Results of four assimilation scenarios with different assimilation periods are given. Columns represent different assimilation scenarios in each year. The red solid line in each subplot represents MLDAS simulation and prediction; the blue dashed line in each subplot represents OL simulation; the green hollow scatters represent SM benchmarks; the blue bars are precipitation; and the shaded regions are the assimilation period, which starts from 1 May and lasts for 15 days, 45 days, 75 days, and 105 days for scenario I, II, III, and IV, respectively.

### 3.1.3. Sensible Heat Flux

The performance of H from OL and MLDAS with a prediction lead time of 30 days is shown in Figure 4. Table 2 shows the $NIC_{RMSD}$ of MLDAS and OL prediction in 2003 to 2005. Overall, $RMSD_{Analysis}$ is decreased compared to $RMSD_{Model}$ in 10 of the 12 experiments. The result indicates that data assimilation can potentially improve the prediction skill of H in the Noah-MP model. Larger $NIC_{RMSD}$ can be seen in scenario IV (assimilation of 105 days) with 57.07% in 2003, 47.80% in 2004, and 47.57% in 2005, indicating that a longer assimilation period results in better prediction performance for H. On the contrary, $NIC_{RMSD}$ is negative in scenario I over 2003 and 2005, revealing that assimilating observations with large uncertainties into the model does not improve the prediction performance, but even degrades it.

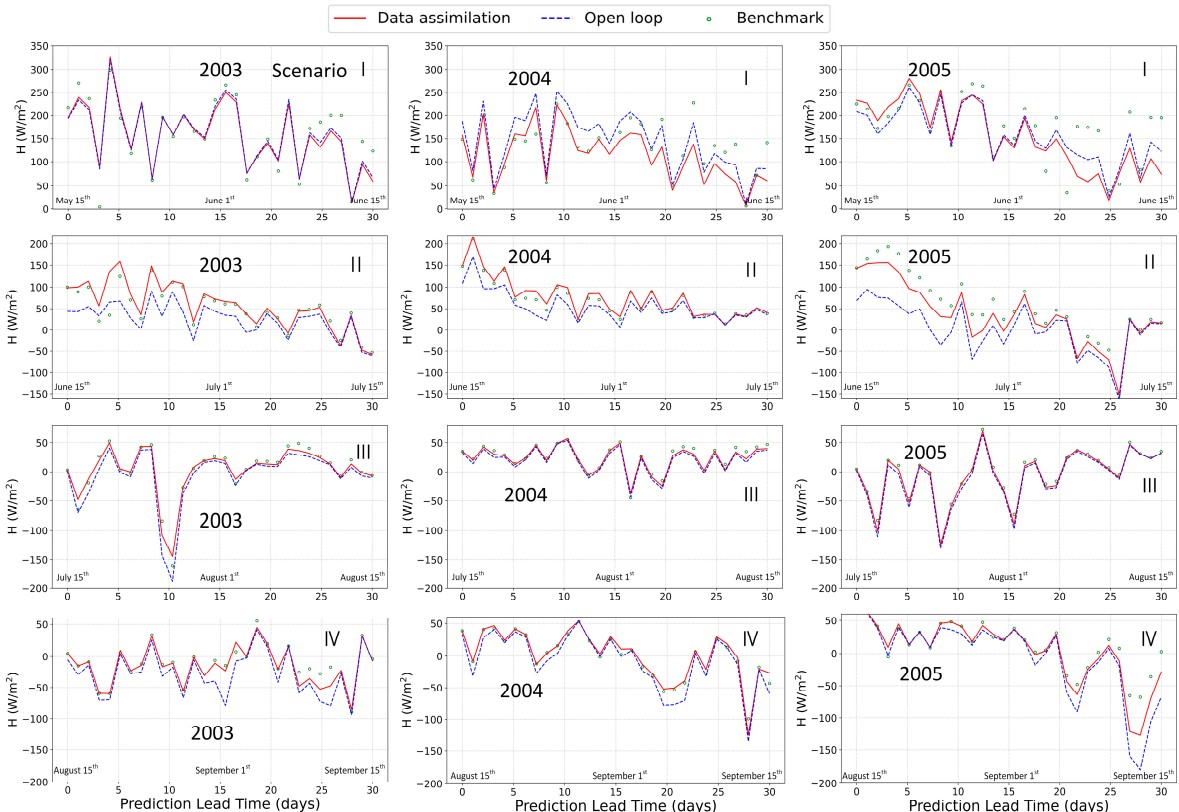

**Figure 4.** Sensible heat fluxes prediction from OL and MLDAS in 2003–2005. Predictions of four assimilation scenarios are given. Columns represent different assimilation scenarios in each year. The red solid line in each subplot represents MLDAS prediction in one future month; the blue dashed line in each subplot represents OL simulation; and the green hollow scatters represent H benchmarks.

**Table 2.** $NIC_{RMSD}$ of sensible heat fluxes in different assimilation scenarios (I–IV) in 2003, 2004, and 2005. Positive $NIC_{RMSD}$ numbers are marked in bold.

| Year | 2003 | | | 2004 | | | 2005 | | |
|---|---|---|---|---|---|---|---|---|---|
| | $RMSD_{Model}$ (W/m²) | $RMSD_{Analysis}$ (W/m²) | $NIC_{RMSD}$ (%) | $RMSD_{Model}$ (W/m²) | $RMSD_{Analysis}$ (W/m²) | $NIC_{RMSD}$ (%) | $RMSD_{Model}$ (W/m²) | $RMSD_{Analysis}$ (W/m²) | $NIC_{RMSD}$ (%) |
| **Scenario I** | 26.716 | 29.411 | −10.09 | 48.168 | 46.886 | **2.66** | 36.926 | 52.082 | −41.04 |
| **Scenario II** | 32.043 | 21.718 | **32.22** | 20.031 | 7.822 | **60.95** | 63.834 | 25.284 | **60.39** |
| **Scenario III** | 15.021 | 7.552 | **49.72** | 7.961 | 5.210 | **34.56** | 9.853 | 5.745 | **41.70** |
| **Scenario IV** | 22.453 | 9.640 | **57.07** | 12.422 | 6.484 | **47.80** | 34.707 | 18.198 | **47.57** |

Details on H prediction in different assimilation scenarios from 2003 to 2005 are given in Figure 4. Considering that there are subtler differences in H prediction between MLDAS

and OL compared with SM, H prediction from four scenarios with lead times of 30 days are given to highlight the differences. In general, both MLDAS and OL can fit the benchmarks quite well. In Scenario I, the OL slightly outperformed the MLDAS in prediction, especially in 2003 and 2005. However, for scenarios II–IV, prediction in MLDAS can better capture the variations in benchmarks both in trend and magnitude. Among scenarios II–IV, scenario II (assimilation of 45 days) improves the prediction skill of MLDAS most significantly, which can also be seen from $NIC_{RMSD}$ in Table 2. Note that H predictions are relatively low with some cases lower than $-50$ W/m$^2$ in scenario II–IV. To confirm the accuracy of benchmarks and predictions, raw observations of H from AmeriFlux were checked. There are also several H observations smaller than 0 W/m$^2$ at this site during the middle of summer, indicating that the benchmarks and predictions are reliable here. Details regarding the observations can be seen at: https://ameriflux.lbl.gov/data/download-data/ (accessed on 10 October 2022). There are probably two reasons for this. Firstly, it is probably because SM simulations are generally higher than the benchmarks (as shown in Figure 3), making H prediction lower than expected. Secondly, irrigation at this site in the middle of the growing season may also be a factor influencing simulation on H. Moreover, the H and LE are mainly affected by the atmospheric state variables (i.e., air temperature and specific humidity) in wet and/or densely vegetated sites [25,26], not land surface variables (e.g., LAI, soil moisture). As it is very humid for scenario III (from Mid-July to Mid-August), the differences between OL and MLDAS will be small.

### 3.1.4. Latent Heat Flux

Statistical evaluation metrics (i.e., $RMSD_{Analysis}$, $RMSD_{Model}$, $NIC_{RMSD}$) of LE predictions are summarized in Table 3. In general, data assimilation significantly improves the prediction skill of LE, and 10 out of 12 experiments obtain positive $NIC_{RMSD}$. Similarly, the MLDAS prediction is slightly worse compared to OL in scenario I (assimilation for 15 days), with a negative $NIC_{RMSD}$ for 2003 and 2005. The prediction performance in MLDAS is improved rapidly in scenarios II–IV, which is similar than that of H. As for the trend of $NIC_{RMSD}$, $NIC_{RMSD}$ increased from 25.69% to 58.08% with a longer assimilation period in 2003. However, the highest $NIC_{RMSD}$ falls in scenario II in 2004 and 2005 (57.34% and 61.56%, respectively), which is similar than the trend of $NIC_{RMSD}$ in H.

**Table 3.** $NIC_{RMSD}$ of latent heat fluxes in different assimilation scenarios (I–IV) in 2003, 2004, and 2005. Positive $NIC_{RMSD}$ numbers are marked in bold.

| Year | 2003 | | | 2004 | | | 2005 | | |
|------|------|------|------|------|------|------|------|------|------|
| | $RMSD_{Model}$ (W/m$^2$) | $RMSD_{Analysis}$ (W/m$^2$) | $NIC_{RMSD}$ (%) | $RMSD_{Model}$ (W/m$^2$) | $RMSD_{Analysis}$ (W/m$^2$) | $NIC_{RMSD}$ (%) | $RMSD_{Model}$ (W/m$^2$) | $RMSD_{Analysis}$ (W/m$^2$) | $NIC_{RMSD}$ (%) |
| Scenario I | 31.121 | 33.785 | −8.56 | 56.808 | 54.251 | **4.50** | 44.714 | 59.053 | −32.07 |
| Scenario II | 36.858 | 27.390 | **25.69** | 22.309 | 9.517 | **57.34** | 69.300 | 26.642 | **61.56** |
| Scenario III | 21.382 | 11.165 | **47.79** | 11.749 | 7.763 | **33.93** | 13.545 | 8.009 | **40.87** |
| Scenario IV | 32.459 | 13.608 | **58.08** | 17.408 | 9.235 | **46.95** | 39.647 | 21.527 | **45.70** |

Figure 5 shows the comparison of the LE prediction with a lead time of 30 days between MLDAS and OL. Prediction in MLDAS, especially in scenarios II and IV of each year, is closer to the benchmarks in most experiments. The most apparent improvement can be seen in scenario II, where OL tends to overestimate LE while MLDAS corrects it. Moreover, the differences between MLDAS prediction and OL prediction diminish with lead time in scenario II over 3 years, indicating that the prediction skill brought by data assimilation also decreases with forecast lead time.

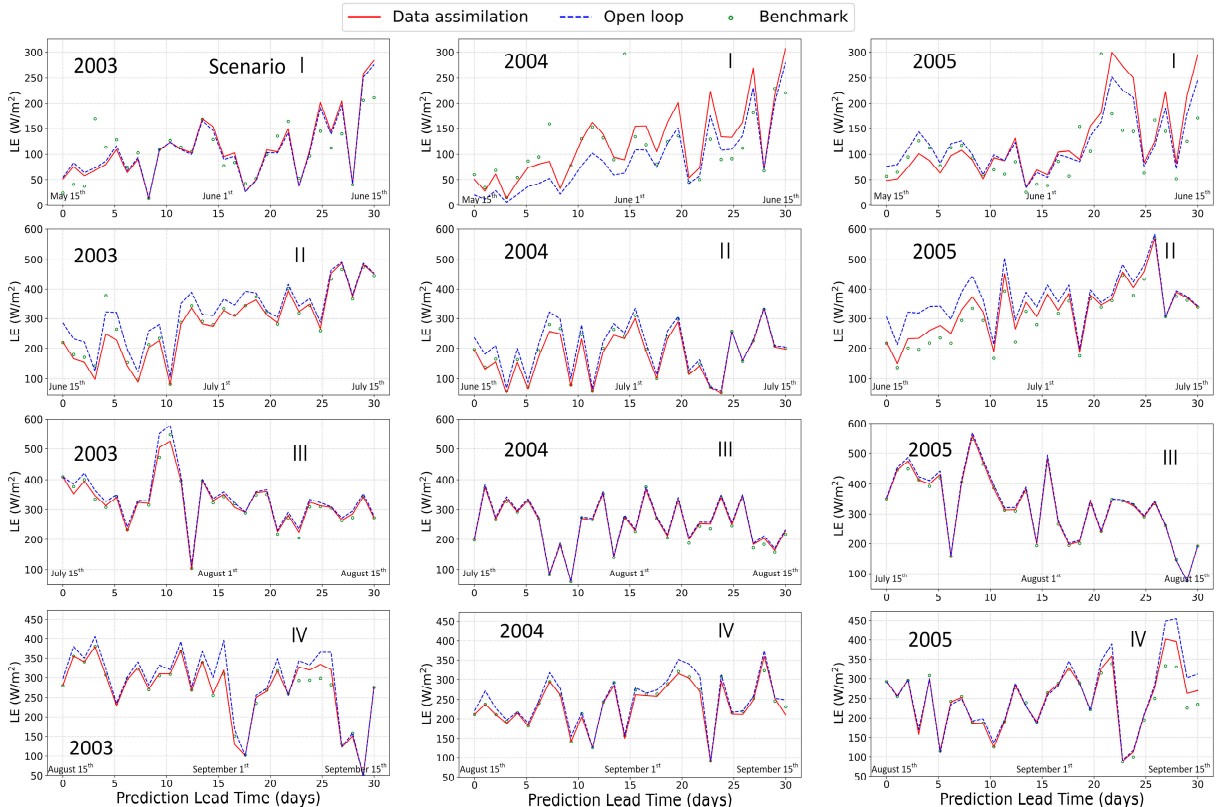

**Figure 5.** Latent heat flux predictions from OL and MLDAS in 2003–2005. Predictions of four assimilation scenarios are given. Columns represent different assimilation scenarios in each year. The red solid line in each subplot represents MLDAS prediction in one future month; the blue dashed line in each subplot represents OL simulation; and the green hollow scatters represent LE benchmarks.

### 3.2. Duration of Impact of Data Assimilation on Prediction

This work focuses not only on whether data assimilation improves the performance of prediction, but also on how long the effect exists. Previous results have demonstrated that MLDAS does have the potential to improve the prediction skill of Noah-MP compared to OL, and specific comparisons are summarized in Tables 1–3. Therefore, it is necessary to analyze the impact of assimilation on the duration of the prediction. Herein, SM and LE are chosen to demonstrate this issue.

The overall dynamic changes of *RD* over four scenarios from 2003 to 2005 are given in Figures 6 and 7. The *RD* of SM prediction is given in Figure 6. As indicated, *RD* in each of the four scenarios decreases with the forecast lead time, suggesting that improved prediction skill brought by data assimilation degrades with time. *RD* in scenario IV is generally larger than other scenarios and the decreasing tendency is not as clear as other scenarios. Figure 7 shows how *RD* of LE prediction changes over forecast lead time. Generally, the trend that *RD* diminishes with lead time is clear in scenarios I and II over the 3 years. Minimum fluctuations of *RD* can be seen in scenario III (assimilation for 75 days), where *RD* is less than 10% and close to 0% over most of the lead time, and corresponding changes can also be seen in scenario III of Figure 6. No significant trend of decreasing *RD* was found in scenario IV, indicating that the impact of data assimilation in scenario IV may persist for a longer period.

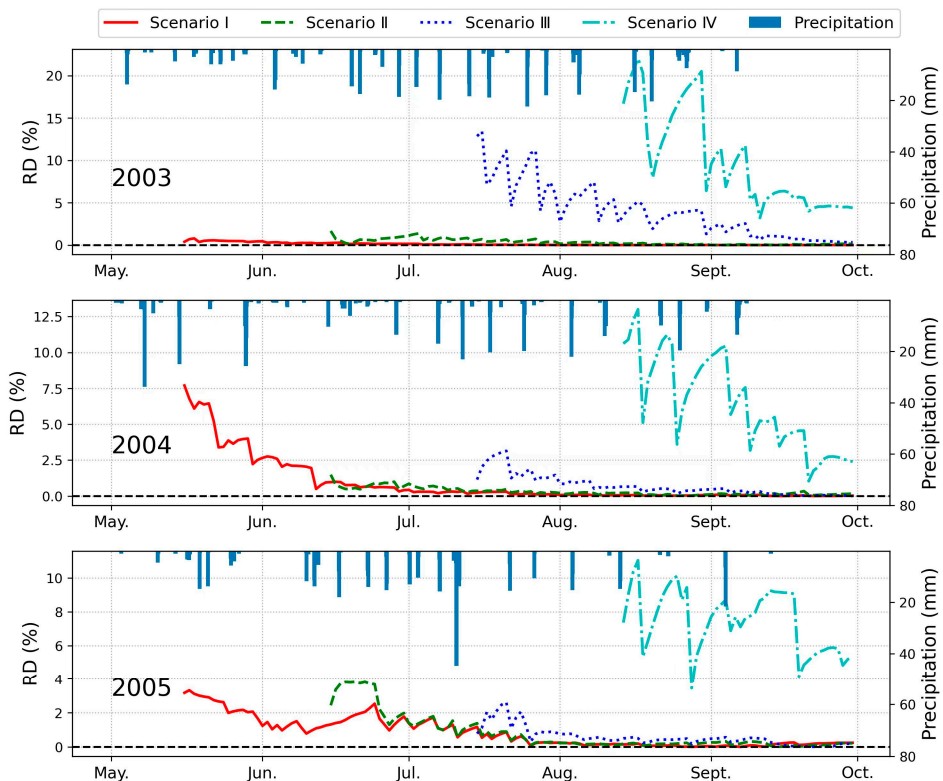

**Figure 6.** *RD* of soil moisture prediction in four scenarios over 2003–2005. The red solid lines, green dashed lines, blue dotted lines, and cyan dashed lines represent *RD* of scenario I, II, III, and IV, respectively. Blue bars are precipitation.

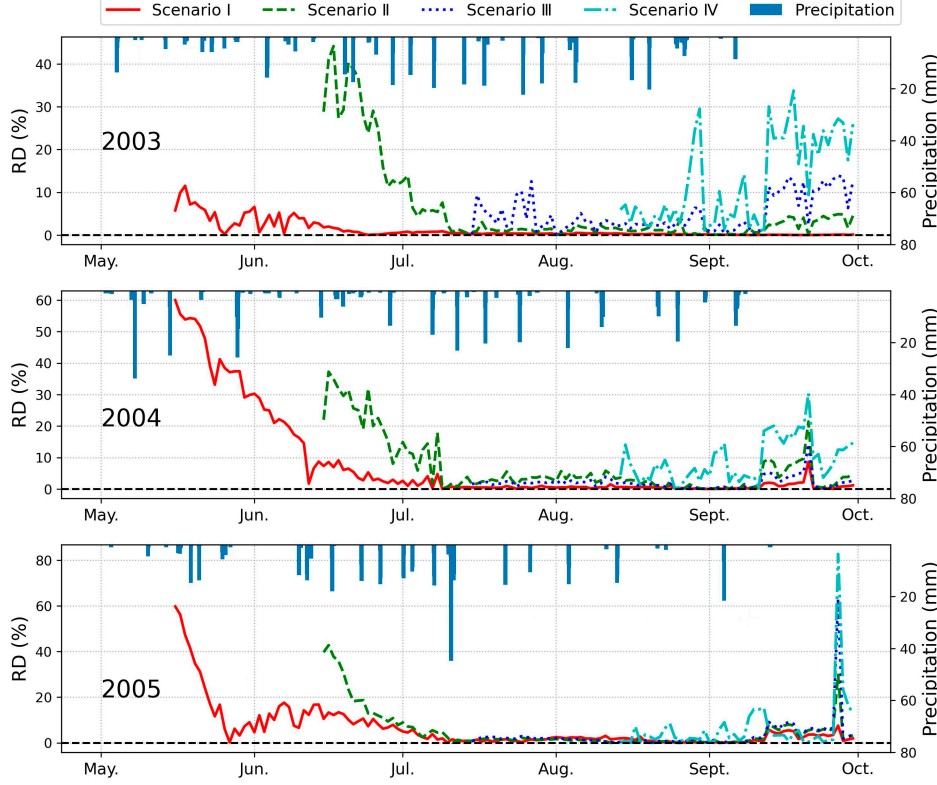

**Figure 7.** Same as Figure 6, but for latent heat flux prediction in four scenarios over 2003–2005.

Scenario I (assimilation for 15 days) of each year (2003, 2004, and 2005) is chosen to show how *RD* changes with forecast lead time so that the forecast lead time can be long enough for evaluation. Figure 8 shows the *RD* of SM over the 3 years. Overall, the influence of assimilation on prediction also diminishes with forecast lead days. The impact of assimilation lasted for around 80 days, 70 days, and 100 days in 2003, 2004, and 2005, respectively. Then the *RD* converges to around 0. Additionally, precipitation is also an essential factor that affects SM simulation. There is a good correspondence between precipitation events and fluctuation of *RD* between MLDAS and OL before convergence (as shown in the grey parts). As is shown, *RD* between MLDAS and OL becomes smaller when precipitation happens, and *RD* increases fast without precipitation (i.e., 20–40 days in 2005), confirming that accurate precipitation inputs are important for SM prediction.

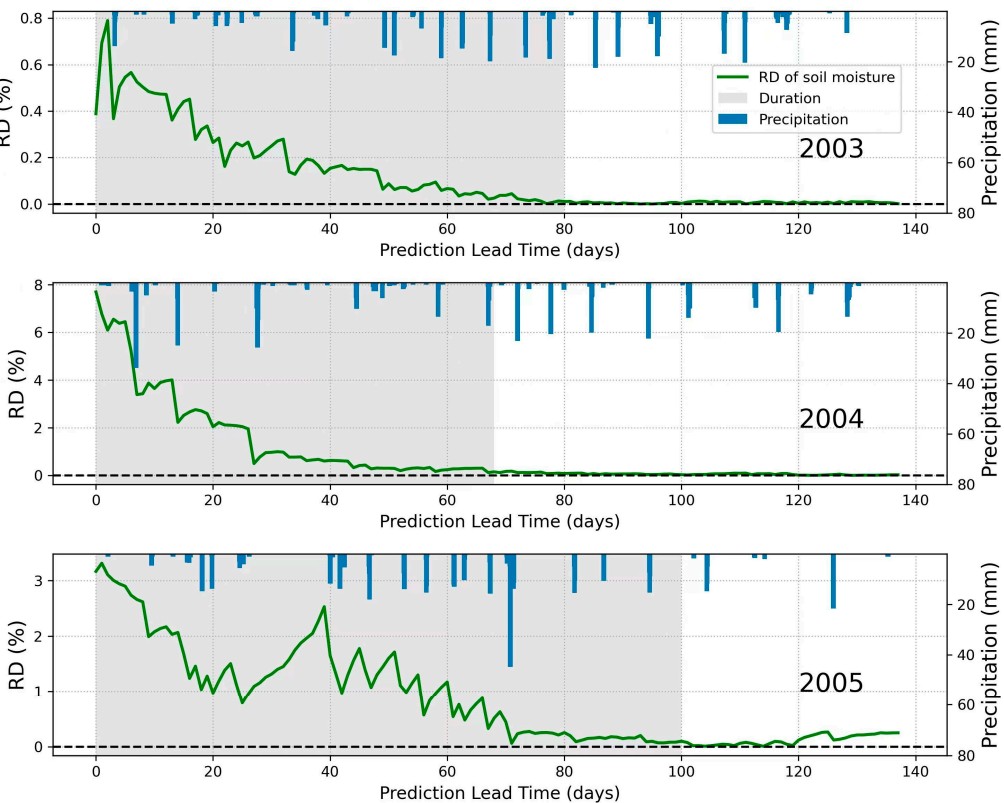

**Figure 8.** *RD* between OL and MLDAS of soil moisture prediction in scenario I in 2003, 2004, and 2005. The shaded area represents the duration of influence brought by assimilation on future predictions.

Figure 9 shows detailed *RD* of LE prediction between MLDAS and OL. Similar to SM prediction, the overall tendency of *RD* diminishes with lead time and apparent fluctuation can be seen in the first 20 to 30 forecast lead days. The *RD* also converged to around 0 at around 60, 55, and 55 lead days in 2003, 2004, and 2005, respectively.

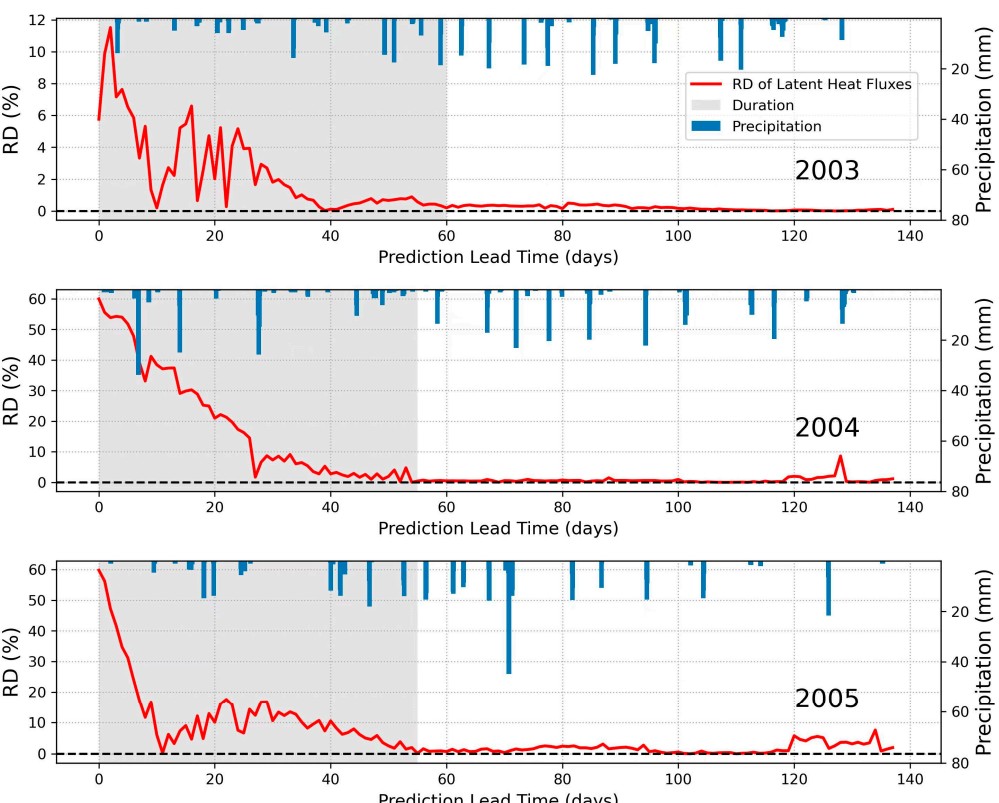

**Figure 9.** *RD* between OL and MLDAS of latent heat flux prediction in scenario I in 2003, 2004, and 2005. The shaded area represents the duration of influence brought by assimilation on future predictions.

## 4. Discussion

### 4.1. Can MLDAS Improve the near Future Prediction Performance?

Most $NIC_{RMSD}$ of H, LE, and SM are positive within four data assimilation scenarios, indicating that MLDAS outperforms OL during the prediction period. It is widely accepted that forecasting errors arise from the uncertainty of initial conditions, model structure and parameters, interactions between land surface and atmosphere, and atmospheric forcing. Previous studies have found that accurate initial conditions are essential for better predictions of land surface variables in LSM [7,27,28]. Wood and Lettenmaier [7] evaluated the relative importance of initial conditions and meteorological forcing in seasonal hydrological forecast and found that initial conditions were more important than meteorological forcing. Here, the initial conditions provided by process-based models (MLDAS or OL) are crucial for future prediction as they determine where the prediction begins. In particular, the forcing data used in our study are from AmeriFlux observations instead of forecasted atmospheric reanalysis, making it possible to highlight the influence of initial conditions on prediction. However, given that the forcing data used here are AmeriFlux observations with high accuracy, the impact of initial conditions may become overestimated and uncertainties in forcing data may not be well represented. Here, for a point-scale simulation driven by AmeriFlux observations, the prediction was likely to be more accurate if the initial condition was close to the "benchmark". For instance, in Figure 3, the initial conditions of SM prediction are different between MLDAS and OL in assimilation-prediction scenario I (assimilation for 15 days) in 2004 (DOY 136, the edge of the grey region), with SM provided by MLDAS (red solid line) closer to the "benchmark". Accordingly, $NIC_{RMSD}$ is 50.51%, indicating that assimilation (assimilation for 15 days) can improve the performance of prediction, despite being rare in scenario I. Hence, initial conditions play an essential role in the performance of prediction [18].

Due to the climatic characteristics of the site, though overall prediction skill of SM, H, and LE is increased in MLDAS compared with that of OL, prediction performance in

different assimilation scenario varies. For instance, as mentioned above, the prediction performance of scenario I is not as good as expected and even degrades in some experiments such as SM, H, and LE prediction in 2005. For the other three scenarios (especially scenario IV) MLDAS received better responses in most experiments. The seasonal transition of land surface variables such as LAI may influence the prediction skill in different assimilation scenarios. As in scenario I, initial conditions are in the spring/summer transition with LAI around 0, which can be seen in Figure 2. Such a low LAI is likely to bring high uncertainties to data assimilation, which lowers the accuracy and reliability of initial conditions and worsens the prediction skill of MLDAS. However, as in scenarios II to IV, the prediction experiments are conducted during the summer or summer/autumn transition with relatively higher LAI. Furthermore, assimilation of SIF in pass IV updates Vcmax25, which is by default a fixed parameter depending on vegetation type in Noah-MP, leading to better LAI simulation. Hence, LAI is assimilated effectively, and lower uncertainty is introduced so that initial conditions are more reliable, contributing to better prediction performance of MLDAS. Regional climate and corresponding phenology are crucial in the forecast framework dominated by initial conditions [7,18], so we suggest that initial conditions for prediction should be carefully checked, considering the uncertainties introduced from climate variations of assimilated land surface variables.

### 4.2. How Long Does the Influence of Assimilation Exist?

Overall, as Figures 6–9 show, the *RD* between OL and MLDAS diminishes and converges to around 0 with forecast lead time, suggesting that assimilation exerts significant influence on prediction during the early prediction period. A similar pattern can be seen in other forecast work with LSM [6]. Relatively accurate initial conditions from data assimilation play an important role during the early prediction period, while the prediction skill brought by initial conditions decreases with lead time due to uncertainties in forcing data. As mentioned in Section 4.1, uncertainties in forcing data are not well considered here. However, the relatively 'perfect' forcing data makes it possible to emphasize the impact of initial conditions on prediction. It can be assumed that the influence of initial conditions on prediction decreases with lead time and disappears when *RD* converges to around 0.

As with fluxes prediction, the impact of assimilation can last for around 60 days. Compared with fluxes prediction, the impact of assimilation on SM prediction generally lasts longer, which can reach around 100 days. This can be explained through the fact that turbulent fluxes are not only controlled by assimilated SM, LAI, and SIF, but also by atmospheric conditions such as advection, condensation, and evaporation during the growing season [29]. Using the same set of AmeriFlux observations as forcing data, it can be considered that the influence of initial conditions for LE prediction is not as significant as that in SM. Furthermore, in the model, LE is updated indirectly by assimilating LAI, SM, and SIF, causing the impact of assimilation on turbulent fluxes prediction to last a shorter amount of time than that of SM. Several studies have also shown that initial conditions of SM significantly impact weather forecast skills in the short or medium range [30–32]. Using AmeriFlux observations as forcing data, uncertainties introduced from precipitation and radiation was not well represented in this study. Coupling WRF to test the forecast skill of MLDAS more comprehensively is needed in the future.

### 5. Conclusions

In this study, four data assimilation scenarios were adapted to evaluate whether the assimilation of SM, LAI, and SIF can jointly increase the prediction performance of MLDAS. Data assimilation experiments were conducted during the growing season over 3 years at an AmeriFlux site. With more accurate initial conditions provided by data assimilation, predictions from MLDAS outperform those from OL. On average, MLDAS produces 28.65%, 27.79%, and 19.15% lower root square deviations (RMSD) for daily H, LE, and SM prediction compared to OL. Moreover, the results also indicate that the impact of assimilation on SM prediction can last longer (around 100 days) than that of LE (around 60 days). Our work

proves that data assimilation can potentially increase the short-term prediction skill of LSMs. Considering that our assimilation-prediction test was conducted at only one site, more sites with different climates and land cover conditions need to be included in our assimilation-prediction experiments to draw some more general conclusions in the future. Furthermore, Noah-MP should be further coupled with atmospheric models (e.g., WRF) to test whether data assimilation influences meteorological forcing, which can give feedback on real-time land surface variable forecasts.

**Author Contributions:** Conceptualization, T.X.; methodology, X.H.; software, Y.T. and X.H.; validation, T.X., F.C., S.L. and X.H.; formal analysis, Y.T.; writing—original draft preparation, Y.T.; writing—review and editing, all; visualization, Y.T.; supervision, T.X.; project administration, T.X.; funding acquisition, T.X. All authors have read and agreed to the published version of the manuscript.

**Funding:** This work was funded by the National Natural Science Foundation of China (42171315) and supported by the State Key Laboratory of Earth Surface Processes and Resource Ecology (2021-ZD-04).

**Data Availability Statement:** Meteorological forcing data of the Mead site including air temperature, pressure, humidity, precipitation, wind speed, incoming shortwave and longwave radiation were downloaded from AmeriFlux (http://ameriflux.lbl.gov/ accessed on 7 October 2021). LAI and SM are also acquired from the AmeriFlux database. The remotely sensed solar-induced chlorophyll fluorescence (SIF) products were downloaded from: https://osf.io/8xqy6/?view_only=4d14a64b7 be644d-39564c82a371e1d20 (accessed on 1 November 2021).

**Acknowledgments:** We would like to thank the academic editor and three anonymous reviewers for their kind help and valuable comments that greatly improved this manuscript. We also would like to thank the high-performance computing support from the Center for Geodata and Analysis, Faculty of Geographical Science, Beijing Normal University.

**Conflicts of Interest:** The authors declare no conflict of interest.

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
