# Peer review of "Can Data Assimilation Improve Short-Term Prediction of Land Surface Variables?"

_remotesensing, doi:10.3390/rs14205172_

Round 1

Reviewer 1 Report

The manuscript use Noah-MP and Multi-pass Land Data Assimilation Scheme to research the short term forecast impacts of SM, LAI and SIF on simulated H, LE and surface soil moisture. It seems that nearly all the research use in situ measurements except SIF products, which fall into the scope of remote sensing. However, the contents about the influence of SIF products are not clearly introduced. I m just wandering whether such research fit for the Journal of Remote sensing. There are also two major concerns which needs to be addressed.

(1) Noah-MP is a land surface schemes, which needs meteorological variables as driving data, I just wander whether it is suitable to use forecast for a land surface model? The data assimilation introduces ground truth information, which will definatelly improve the simulation results. But it is still not clear how to do multi-parameter data simulation for SM, LAI and SIF? How could the simulated information, especially SIF, influence the future soil moisture, H and LE?

(2) There are definitely differences exist for simulation of SM, H and LE, when additional land surface variables are included in the estimation. In the whole manuscript, there needs additional efforts to make clear why data assimilation can improve or degrades the simulation, compared with the in situ measurements. For example, The NoahMP model may overestimate or underestimate the sensible heat flux or latent heat flux just because of the biased land surface parameterization scheme. Also there are many other factors which can influence the simulated heat fluxes. Could you explain why in Scenario III, why the two simulations nearly have no differences?

(3) Line 351, remove to in the sentence; the figure quality needs to be improved. 

Reviewer 2 Report

In this article the authors examine the impact of land surface data assimilation on the forecasts quality of land surface variables such as LAI and soil moisture, together with surface turbulent fluxes.  This study is relevant and of interest, but a few major modifications are required in my opinion before it should be considered for publication in Remote Sensing. 

Major issues:

First major issue:  The results presented in this study are based on a single site.  At the very least, the authors need to clearly acknowledge this fact and mention this as a limitation of their study, and be careful not to be too categorical with their conclusions (see examples of that at the end of the minor issue section). 

Second major issue:  The term “forecast” is not correctly used in this study.  The model runs referred to as “forecasts” are in fact forced with observations from the Ameriflux Mead site.  A large part of the error and uncertainty for real forecasts comes from errors associated with incoming radiation and precipitation, which are under represented in this study.  Several of the conclusions are likely to be too optimistic because this component is not taken into account.    

Third major issue:  In Figure 4, values for sensible heat fluxes seem unrealistically small.  For example, there are values close to zero and even negative for 2004 and 2005 for scenarios II, III, and IV, which does not make sense during summertime, even if the soils are quite humid as they are over that area.  The authors need to explain that.  The trend (decrease with time) also does not seem right, even if these fields are irrigated (are they?)

Minor comments:

Line 38: “… While LSMs can provide temporally and spatially continuous simulations of land surface variables, there are still substantial uncertainties caused by initial conditions, forcing data and model physics within LSMs…”.  I would also add the interactions between the land surface and the atmosphere, which is also highly uncertain.   

References:  there are two articles listed as number “9” in that section.

Section 2.1:  Please provide more information about the Mead Ameriflux site. 

Line 117:  “…Currently, four passes have been built to assimilate LAI (pass I – pass II), SM (pass III), and SIF (pass IV) to update specific leaf area (SLA), leaf biomass (LFMASS), SM and maximum rate of carboxylation (Vcmax) in MLDAS successfully [18]. LAI observations will be assimilated when available, and SM and SIF observations will be assimilated when available at daily and four-day intervals, respectively.”  Maybe the authors could explicitly list the control variables for the multi-pass data assimilation.  It includes soil moisture, but does it also include soil / surface temperature?  Vcmax is a parameter, not a prognostic variables, is that right? Should it be included as a control variable?  Is the vegetation fractional coverage also a control variable, and LAI?

Section 2.4:  More details are necessary in that section.  Is the evaluation done for the same month?  (Does not look like this is the case.)  Are the atmospheric forcing the same for the open loop, data assimilation, and forecasts?

Line 137:  “A positive NICRMSD indicates the forecast skill of MLDAS is improved compared with OL. A negative NICRMSD means degradation.”  If that’s true, then Equation (1) is not written correctly.  What the current formulation, lower values of RMSDAnalysis will lead to negative values of NICRMSD, not positive. 

Figure 1:  Should also include a schematic of the benchmark and of the open loop in that figure.  Also, based on that figure, the forecasts are only for one month, but results presented later suggest that the “forecasts” were longer than that.

Section 3.1.1:  Maybe the authors could add some texts (in that section or in the discussion) about the predictability of LAI.  The results suggest that just one month of DA early in the season is sufficient for a good prediction of this quantity.  Longer DA does improve much after that. 

Figure 2: (and other figures).  I would like to have more details in the figure caption.  E.g., description of the different lines, symbols, units, shading, validity period, etc…

General comment:  why use data from 2003 to 2005, and not something more recent?

Line 172:  “…that forecasting performance may get optimized robustly with…” Maybe rephrase that part, “may get optimized robustly”. 

Table 1. Please indicate the units.  Also, are the results for one month, or for longer?  If it’s for one-month forecasts, then the period is not the same and comparison is difficult. 

Table 1.  Why does the RMSD increases with the scenario number?  Wouldn’t we expect the opposite?

Table 1.  The RMSD values seem quite small for soil moisture forecasts (even analyses), compared with other studies.  Maybe the authors could comment on that?  This probably related to the fact that this is a comparison with a model benchmark. 

Table 2. Same comments as for Table 1 (units, small values of RMSD)

Line 238:  “Herein, SM and LE are chosen and mean absolute percentage error (MAPE) between forecast of MLDAS and OL are used to demonstrate this issue.”  This should be better explained in the text.  I did not understand it the first time I read it, and only when I looked at the figures did I see that this is a measure that provides the gain (in percentage) between MLDAS and OL.  In fact, it could be named differently, to make it clearer that this is a difference between two experiments.    

Figures 6 and 7:  … just to say that I like these figures very much.  Quite interesting.  Just to be sure, we are looking at the forecasts here, right?

Line 267:  “… implying that accurate precipitation inputs are important for SM forecast…” Well, this is certainly well know.  Maybe change “implying” by “confirming”. 

Figure 8, caption:  explain the meaning of the shading.  Could be confused with the assimilation period (as is done in some of the other figures).    

Figure 9:  The top and bottom panels show a rather strange behavior of MAPE at the left end side (early in the period examined).  We first see a rapid decline of MAPE until day 10, followed by an increase and decrease all the way to day 50-60.  Can the authors provide some explanations for that MAPE time evolution?

Line 281: “that MLDAS outperforms during…” outperforms what?

Line 296: “To a large extent, the optimization of initial conditions provided by data assimilation improves the short-term forecast performance of LSM.”  This conclusion is a bit of a stretch because you don't really have forecasts here... the atmospheric forcing and the model processes would have a bigger impact. 

Line 323: “It can be assumed that the influence of initial conditions on forecast disappears, and meteorological forcing becomes the dominant factor when MAPE converges to around 0.”  Actually this transition would occur much faster in “real life”, where the uncertainty associated with the atmospheric forcing (and other factors) is much larger and dominating. 

Line 330:  “Under such circumstances, the tradeoff between initial conditions and meteorological forcing in fluxes forecast is different from that of SM. That is, meteorological forcing is tended to play a more important role than initial conditions with lead time, causing the impact of assimilation on turbulent fluxes forecast lasts shorter than that of SM. Compared with fluxes closely related to atmospheric conditions…”  The authors should be careful with this conclusion, since the atmospheric forcing they use in this study can be considered as “perfect”, and their impact is not representative of what would occur for real-life forecasts, with much greater uncertainty for the forcing. 

Line 336:  “Several studies have shown that initial conditions of SM have significant impact on weather forecast skills in the short or medium range. Thus, initial condition exerts more influence to forecast skill for SM than meteorological forcing and a relatively more accurate initial condition of SM through data assimilation can optimize the forecast for a longer period.”  Can you provide references for that first sentence?  I disagree with that second sentence… at the very least the authors could be more circumspect with this last conclusion and show some restraint. 

General comment:  Do the authors know what type of observations has the greatest contribution to the quality of the forecasts, i.e., soil moisture vs LAI vs SIF?  In fact, what is the role / impact of SIF in this study?

General comment:  why not use actual measurements of soil moisture and turbulent fluxes for the evaluation? 

Line 353:  “… draw some more universal conclusions… “ I would use “general” instead of “universal”.

Line 354:  “LSMs should be further coupled with atmospheric models (e.g., WRF) to test whether data assimilation influences meteorological forcing, which can give feedback on real-time land surface variable forecasts.”  This is done indeed (at least with other atmospheric models), and results indicate that the accuracy of soil moisture forecasts degrade rather quickly due to large uncertainty for the atmospheric forcing (incoming radiation, precipitation). 

Reviewer 3 Report

In this article the abstract is well developed, including relevant information about the main aspects.

It is not clear from the introduction why the article is so outdated in terms of analyzed period (17 - 19 years ago). In addition, there is only a small number of international references introduced in the text.

Figures 6 and 7 can be improved in terms of quality.

Overall, the article is well structured, the quality of presentation is good and the content can present interest for readers.

However, it should be clarified why the authors selected 2003 - 2005 as studied period (why not a period from a recent past, why only 3 years etc.).

Round 2

Reviewer 2 Report

I would first like to thank the authors for the many changes they have made to the manuscript following my first round of comments.  I find this version more acceptable for publication in Remote Sensing.  But there are still a few items that are problematic and for which I wish the authors could further modify their submission.

About the use of the term “forecast”:  Even though the authors now better explain how their model runs are generated, they are still using the term “forecasts”.  As a reminder, forecasts are prediction of the future state of a system, based on past and current data… that is, they cannot use information from the future such as what is done in this study with the atmospheric forcing.  I strongly suggest the authors to find another term for the model runs, which are in fact open loops with modified initial conditions. 

Line 245: “Note that H forecasts are relatively low with some cases lower than -50W/m2 in scenario â…¡-â…£. It is probably because SM simulation are generally higher than benchmarks (as shown in Figure 3), making H forecast lower than expected. In addition, irrigation at this site in the middle of growing season may also be a factor influencing simulation on H. Moreover, the H and LE are mainly affected by the atmospheric state variables (i.e., air temperature and specific humidity) in wet and/or densely vegetated sites [25,26], not land surface variables (e.g. LAI, soil moisture).”  I agree that irrigation could explain why sensible heat fluxes have relatively lower values… but not values close to “0” or even negative in the middle of summer.  It’s true that atmospheric state variables play an important role for the surface heat fluxes when soils are very wet (because surface temperature does not change much, with a less pronounced diurnal cycle), but negative fluxes implies that the surface is actually cooler than the atmosphere (which is quite outstanding in the middle of summer).  I believe the authors should provide more details on this, and use observed flux observations (available?) to confirm these values. 

Figure 4 caption:  since these runs (forecasts) are produced over different months for the 4 scenarios, it would be good if the authors could indicate the month on the figures.  In the captions at the very least. 

Line 343:  “Specially, the forcing data used in our study are from Ameriflux observations instead of forecasted atmospheric reanalysis, making it possible to highlight the influence of initial conditions on forecast”.  I agree with that statement… but it should also be very clearly stated that there are also disadvantages with using Ameriflux observations as atmospheric forcing, i.e., it should be explicitly mentioned that this study overestimate the impact of initial conditions on predictability of land surface variables.  For instance, if “real” predicted atmospheric forcing are very uncertain, than the impact of improving initial conditions will be substantially smaller as much of the error growth early in the forecasts will be determined by the atmospheric forcing. 

Line 379:  “Considering that the forcing data used for OL and MLDAS are same and of high accuracy, the influence of initial conditions are highlighted. It can be assumed that the influence of initial conditions on forecast decreases with lead time and disappears when RD converges to around 0.”  Same comment here.  I don’t disagree with that statement, but it does not mention the limitations of using Ameriflux observations data as atmospheric forcing for the model runs.  
